# Improving the working environment for the delivery of safe surgical care in the UK: a qualitative cross-sectional analysis

Alice Baggaley,[1] Lydia Robb,[2] Simon Paterson-Brown,[2] Richard J McGregor[2]

¹Department of Surgery, Homerton University Hospital, London, UK
²Clinical Surgery, Edinburgh Royal Infirmary, Edinburgh, UK

**Correspondence to**
Dr Richard J McGregor;
Richard.McGregor@ed.ac.uk

## ABSTRACT

**Objectives** The aim of this study was to identify current problems and potential solutions to improve the working environment for the delivery of safe surgical care in the UK.

**Design** Prospective, questionnaire-based cross-sectional study.

**Setting/participants** Following validation, an electronic questionnaire was distributed to postgraduate local education and training board distribution lists, the Royal College of Surgeons of Edinburgh (RCSEd) mailing lists and trainee organisations. This consisted of a single open-ended question inviting five open-ended responses. Throughout the 13-week study period, the survey was also published on a number of social media platforms.

**Results** A total of 505 responders completed the survey, of which 35% were consultants, 30% foundation doctors, 17% specialty trainees, 11% specialty doctors, 5% core trainees and <1% surgical nurse practitioners. A total of 2238 free-text answers detailed specific actions to improve the working environment. These responses were individually coded and then grouped into nine categories (staff resources, non-staff resources, support, working conditions, communication and team work, systems improvement, patient centred, training and education, and miscellaneous).

**Conclusions** The results of this study have identified a number of key areas that, if addressed, may improve the environment for the delivery of safer surgical care. Common themes that emerged across all grades included: increased front-line staff; a return to a 'firm' structure to improve team continuity; greater senior support; and improved hospital facilities to help staff rest and recuperate. While unlimited funding remains unrealistic, many of the suggestions could be implemented in a cost-neutral fashion and include insightful ideas for remodelling or restructuring the workforce to improve the efficiency of the surgical team. The findings of this study formed the basis of a set of recommendations published by the RCSEd as a discussion paper.

## Strengths and limitations of this study

► This study describes the detailed perspectives and opinions of front line National Health Service staff regarding current problems and potential solutions for improving the delivery of safe surgical care in the UK at a time of unprecedented strain.

► The wide distribution of this survey throughout the UK and response rates across the training/non-training grades helps to mitigate against subgroup bias and generates a depth and richness to the answers.

► While some groups had a higher number of responders than others, this is possibly explained by varying degrees of penetration and distribution across the electronic mailing lists.

► Despite the brevity of the survey, we found a higher than expected incompletion rate (46%); however, the demographics of those who did not fully complete the survey were comparable with those that did complete the survey, eliminating a potential completion bias of the respondents.

► The extended surgical team is becoming ever more important when considering the multidisciplinary approach to the delivery of acute surgical services, and therefore, the limited responses from this group of staff is a weakness.

## INTRODUCTION

Human factors and ergonomics (HF&E) is an independent specialty and profession that lies at the intersection of psychology and engineering, with the goal to design working environments to support human performance and improve safety. One of the key principles of HF&E in healthcare is that healthcare professionals do not make errors in isolation, and the working environment should be adjusted in order to support those working within it in order to mitigate these errors.[1] Increasingly, the expertise of HF&E specialists is being used across the healthcare spectrum, including the domain of surgery, to help us redesign our 'systems' and approaches to improving patient safety.[2 3]

While the health and well-being of the workforce underpins a small piece of the overall specialty of HF&E in healthcare, which in turn ties into patient safety, it is intrinsically linked to team work and team performance.[4] Moreover, there is a vast array of literature outwith healthcare, and increasingly within, which focuses on the critical relationship between non-technical skills, team performance and improved clinical outcome.[5] The intimate link between human error and patient safety

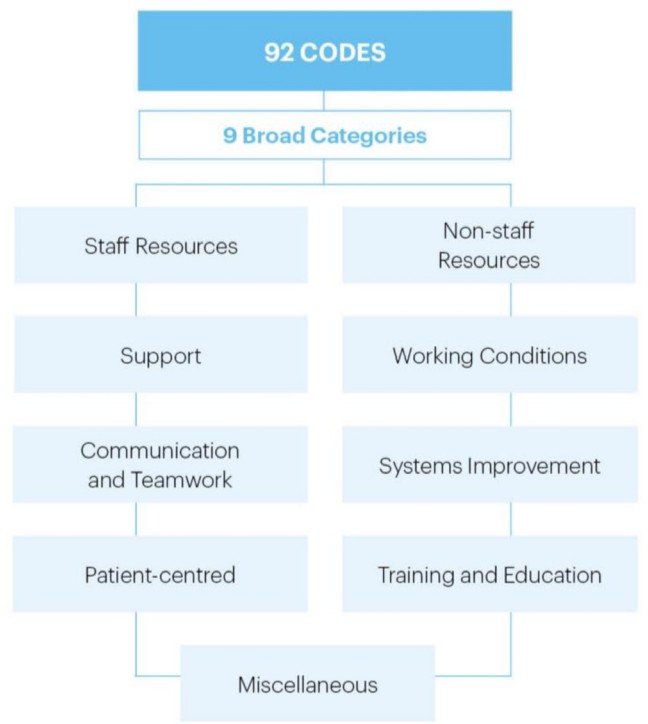

**Figure 1** Nine categories. Schematic breakdown of the nine broad categories the results were grouped into, from a total of 92 codes generated during analysis.

had been placed firmly under the spotlight by a number of key studies that estimate that approximately 10% of all hospital admissions result in an adverse event.[6] This was directly responsible for the introduction of variously titled 'Patient Safety' organisations around the world and the development of 'bundles of care' for standardising protocols for a number of procedures and the WHO's 19-item Surgical Safety Checklist.[7]

The delivery of safe surgical care within the UK is a complex and multifaceted issue. To date, there has been limited direct engagement with the stakeholders on the front line of acute surgical services to ascertain what they feel are the current problems in delivering safe surgical care and the potential solutions. The work of this project was initiated through the creation of a short life working group (SLWG) within the Royal College of Surgeons of Edinburgh (RCSEd). The objective of this group was to identify the fundamental factors that could improve the working environment for all those involved in providing safe surgical care. The SLWG aimed to target surgeons and nurse practitioners who were involved in the day-to-day running of the service. The qualitative data collected was via a cross-sectional survey, from which a number of themes emerged.

## METHODS
### Questionnaire design and distribution
A prospective, cross-sectional, qualitative, online survey was created using 'Survey Monkey'. The survey content was developed by a RCSEd SLWG (the members of which

are detailed in the acknowledgements section). This was approved and validated by the Patient Safety Board, with the explicit aim to elicit as broad a response as possible, from the breadth of the multidisciplinary team.

The survey was distributed to members of the surgical teams in hospitals across the UK, using the RSCEd mailing list, postgraduate local education and training board distribution lists, trainee organisations (eg, Association of Surgeons in Training) and deanery mailing lists. The utilisation of mailing lists and social media platforms accessible to representatives for foundation trainees, core and specialty trainees provided UK-wide distribution.

The email or social media post detailed the purpose of the survey and included a hyperlink that connected directly to the survey on the 'Survey Monkey' website. Email distribution began in February 2016, with all groups receiving a reminder at 6 and 10 weeks. The survey was open for responses for 13 weeks between 8 February to 14 May 2016. Social media engagement was used throughout the aforementioned period.

The survey was entirely voluntary, and no identifiable data were collected or stored. Participants were asked to provide basic demographics including: gender (male or female); grade (consultant, registrar, specialty doctor, core trainee, foundation year trainee, nurse and other); location (England, Wales, Scotland and Northern Ireland); and hospital setting (large city hospital, district general hospital and remote and rural hospital). Participants could select the most relevant from predetermined drop-down options detailed in brackets above.

The questionnaire consisted of one single open-ended question: 'In the delivery of safe surgical care, what are the five most important things which would improve the workplace environment <u>for you</u>?'. Thereafter, sequential white-space boxes allowed for a maximum of five free-text responses (up to 4000 characters each). Survey design dictated mandatory completion of demographics prior to progression onto the free-text responses.

### Data analysis
Free-text responses were individually coded by two of the authors to allow categorisation. For example, this free-text response from a consultant, 'Improve functionality of electronic patient information systems, particularly prescribing, with access device at every bed space specific to that patient to return ward round assessment to what it was 5 years ago', was assigned the code 'Better IT systems/computers/software', which was subsequently grouped into the resources category. A minimum of one free-text response was required to meet inclusion criteria. The final 92 codes were discussed by the SLWG who agreed on grouping of the codes into nine broad themes (figure 1). The nine themes were as follows: staff resources, non-staff resources, support, working conditions, communication and team work, systems improvement, patient centred, training and education, and miscellaneous. Microsoft Excel (Microsoft, 2010, Redmond, Washington, USA) was used to store the data and generate descriptive statistics.

| Recommendation 1 | Establish structured senior support |
|---|---|
| Recommendation 2 | Reintroduce the Hospital Mess |
| Recommendation 3 | Intelligent Design of Rotas |
| Recommendation 4 | Streamline and reorganise the overall workload to prioritise core clinical duties and creat and integrated multidisciplinary surgical team |
| Recommendation 5 | Recognise that better training delivers better care |
| Recommendation 6 | Promote Human Factors training |
| Recommendation 7 | Support and Training the Trainers |

**Figure 2** Summary of RCSEd key recommendations. These seven recommendations were published in an RCSEd discussion paper titled: '*Improving the Working Environment for Safe Surgical Care*', and were proposed in part using the data from this survey. RCSEd, Royal College of Surgeons of Edinburgh.

## Patient involvement

The questionnaire used in this study was edited and approved by the RCSEd Patient Safety Board, which has lay person representation in addition to medical professionals. The Patient Safety Board also helped devise the key recommendations[8] published by RCSEd following the analysis of the survey results (see figure 2).

## RESULTS

A total of 932 people started the survey; however, only 505 responders proceeded to complete at least one free-text response, resulting in a 54% completion rate. Those 427 remaining responders who did not meet the inclusion criteria of entering at least one free-text answer were excluded from further analysis. The 505 participants generated a total of 2238 individual freetext responses. The demographics of the respondents are shown in figure 3.

The greatest number of free-text responses was generated from consultants and foundation year trainees (801 and 651, respectively). Only six nurse practitioners completed any free-text responses, and their results are not presented here due to insufficient numbers when in comparison with other grades. Figure 4 shows the distribution of responses across each of the nine broad categories, regardless of grade. The vast majority of responses relate to resources and were further subdivided as either staff

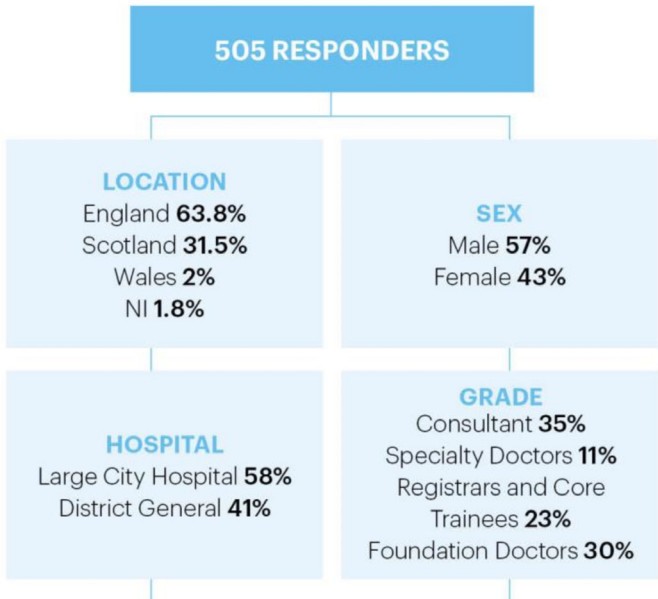

**Figure 3** Demographics of responders. This figure summarises the demographics of those who responded to the survey and met the inclusion criteria (n=505). NI, Northern Ireland.

or non-staff related resources. The next most frequently recurring responses fell into the communication and teamwork, working conditions and support categories.

There were a high number of recurring responses throughout the survey, and figure 5 lists the top five from each grade. The need for more staff was an area of concern identified across all grades. This included more doctors and increased numbers of all members of the surgical team. Interestingly, the need for more administrative and secretarial staff was mentioned regularly, which may represent the high burden of 'paperwork' in the modern working environment.

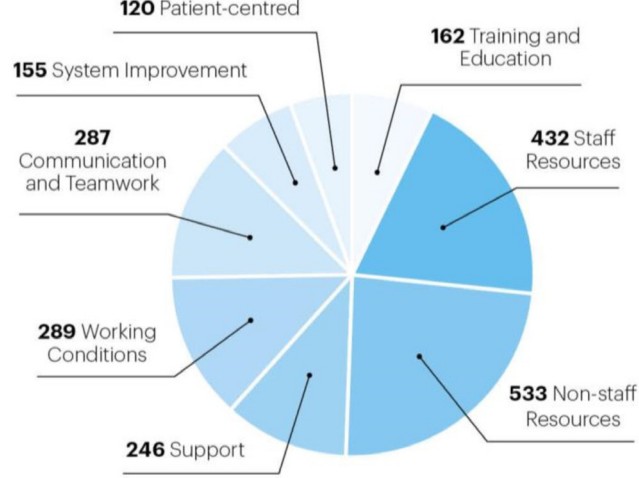

**Figure 4** Responses across main categories. This pie chart demonstrates the total number of responses within each category regardless of grade. With a total of 505 responders, 2238 individual free-text answers were generated.

| Grade | Top 5 Responses |
|---|---|
| Consultant (n= 173) | Increased staffing levels |
| | Return to firm approach |
| | Improved access to imaging/diagnostic services/ interventional radiology |
| | Improved access to emergency theatre |
| | Ring-fenced beds for surgical patients |
| Speciality Registrar (n= 83) | Increased staffing levels |
| | Improved facilities |
| | Return to firm approach |
| | Improved IT systems |
| | Improved handover |
| Core Trainee (n= 27) | Increased staffing levels |
| | Senior support |
| | Support from other specialties |
| | More time in theatre |
| | Improved facilities |
| Specialty Doctor (n= 57) | Increased staffing levels |
| | Senior support |
| | Teaching/training |
| | Adequate equipment |
| | Improved facilities |
| Foundation Doctor (n= 151) | Increased staffing levels |
| | Senior support |
| | Teaching/training |
| | Improved IT systems |
| | Better communication |

**Figure 5** Top five responses according to grade. This table demonstrates the top five coded responses per grade. This figure also details the total number of responders within each grade as n=x, from the total of 505 responders who met the inclusion criteria.

## Consultants

Consultants provided 801 individual free-text responses; over half of these related to resources, both staff and non-staff related (figure 6). The comments regarding non-staff resources included suggestions such as 'better information technology systems' (n=28), 'more beds' (n=28) and 'better availability of imaging/diagnostic services' (n=37). The most frequent individual code pertained to increased staff (n=77), which included medical and surgical axillary staff but particularly noted the need for more nursing staff.

Within the working conditions category, over half of the responses pertained to 'continuity of team members' and 'returning to a "firm" approach' (n=55). For the support category, consultants mostly referred to support from hospital management (n=25), as opposed to that of senior colleagues as suggested by the more junior grades. Within the communication and teamwork category, consultants suggested better communication skills (n=10), improved teamwork (n=19) and improved morale (n=13) to enhance the working environment. For systems improvements, the need for less bureaucracy

| Categories | Consultant Responses (%) | Speciality Doctor Responses (%) | Registrar and Core Trainee Responses (%) | Foundation Doctor Responses (%) |
|---|---|---|---|---|
| Staff Resources | 15 | 13 | 18 | 20 |
| Non-Staff Resources | 37 | 30 | 22 | 14 |
| Support | 6 | 10 | 9 | 18 |
| Working Conditions | 12 | 10 | 17 | 12 |
| Communication and Teamwork | 9 | 7 | 14 | 20 |
| Systems Improvement | 10 | 8 | 5 | 3 |
| Patient Centred | 7 | 5 | 6 | 3 |
| Training and Education | 3 | 16 | 8 | 10 |
| Miscellaneous | 1 | 1 | 1 | 0 |

**Figure 6** Percentage of category response according to grade. This figure details the breakdown of response across the categories within each grade. The grades include consultants, specialty doctor, registrar and core trainees together, as well as foundation doctors. The numbers detailed within this figure are a percentage of the total number of responses within that specific grade.

(n=20), less focus on service provision (n=11) and a reduction in use of protocols was suggested (n=15). For the patient-centred category, the popular responses were continuity of patient care (n=11) and the need to operate on ones' own patients (n=11). This links to the frequently suggested need to return to a 'firm' structure in order to improve continuity of team members.

### Registrars and core trainees
Collectively, this group provided 434 free-text responses, with the largest category being non-staff resources (22% compared with 37% of consultant responses) (figure 6). Within the working conditions category, continuity of team members (n=25) and improvement to the rota (n=12) were both popular codes. However, the most frequent code was the suggestion of improvement to facilities such as a hospital mess, increased parking, office space, changing rooms, showers and out of hours catering (n=17). These suggestions were common across all grades. Registrars felt improvements to handovers were important (n=13)—proportionally more so than any other grade—and this may reflect the training posts that the registrars and core trainees hold. Another common theme was that of improvement to appraisals and feedback, as well as an increase in both quality and quantity of teaching (n=20).

### Specialty and associate specialty doctors
Aligned with the response of the consultant body, around one-third of specialty doctors' responses related to non-staff resources such as adequate equipment (n=17) and improved computer systems (n=8). Of all the grades, specialty doctors provided the highest number of responses within the training and education category (figure 6), which mostly related to career progression and access to, and recognition of, training (n=13). Specialty doctors also sought greater support from their seniors as

well as hospital management (n=9). Subjectively, there appeared to be a sense of frustration at being under-valued within the system. This is of particular importance as progression into training posts declines, making recruitment and retention of specialty doctors even more vital.

### Foundation year doctors
Foundation doctors comprised the second largest group after consultants, with 651 free-text responses generated (figure 6). Compared with the other grade groups, foundation year doctors provided the lowest proportional number of responses regarding non-staff resources but the highest number of responses regarding support (n=120), which frequently involved suggesting more support from their senior colleagues. Compared with consultants, responses from foundation year doctors were much more concerned with communication and teamwork (20% vs 9%) and suggested that improved handovers, better team cohesion and clearer plans from their consultants regarding patients might improve the working environment for safe surgical care.

Within the category staff resources, foundation year doctors provided the highest total number of responses (n=129) and tended to seek more staff to support their ward roles, such as phlebotomists, healthcare assistants and physician associates, as well as more 'juniors' to facilitate the staffing of a full rota. Certainly, the recognition of the increasing load of non-clinical ward tasks has led to expansion of the 'wider surgical team', with the recruitment of advance nurse practitioners, prescribing pharmacists and physician associates.

### DISCUSSION
Many of the themes highlighted by the responses were interconnected and shine a spotlight on the unprecedented strain the National Health Service (NHS) is currently experiencing. For example, it is easy to see how 'rota gaps' left by too few staff can lead to low morale, poor communication and a lack of continuity of care and teamwork. While unlimited funding remains unrealistic, many of the suggestions from front line staff could be implemented in a relatively cost-neutral fashion. For instance, the use of smart rotas that aim to increase daylight training time and return to a firm structure[9] could improve patient safety, morale and training and in turn improve recruitment and retention within surgery. Examples of responders' comments can be seen in figure 7. The main core themes highlighted by the survey are discussed below.

### The need for structured senior support
There was a very strong message from the free-text answers that better supervision and support for both foundation doctors and specialty trainees would improve overall safety. It is postulated, quite reasonably, that better supervision and support would lead to less

| Categories | Example Response |
|---|---|
| Staff Resources | "More value put on support staff such as secretaries and ward clerks to ensure administrative processes work reliably." |
| Non-Staff Resources | "Better out-of-hours access to urgent radiology would improve surgical safety." |
| Support | "We ought to have a senior-led ward round at least once per day." |
| Working Conditions | "There should be on-call facilities so night staff can rest if they are not busy and can sleep in the morning if they are too tired to drive home after night shift." |
| Communication and Teamwork | "Less barriers between senior surgical staff and junior staff : often junior staff can get intimidated and bear the brunt of bad moods which does not allow for safe care of the patient." |
| Systems Improvement | "Consultants to run departments with managers responsible to them - too many clinical decisions are being pushed through by managers who have no knowledge about the consequences if their decisions" |
| Patient Centred | "Junior doctors are always on shifts. They do not provide continuity of care and do not know the patients. the burden is fully on the consultant" |
| Training and Education | "Juniors being given explanations of the surgical procedures which their patients undergo (including having opportunities to go to observe them in theatre where possible), including knowledge of common complications and how to treat them" |

**Figure 7** Representative qualitative comments from respondents. This figure provides an example of a qualitative comment from each of the nine categories. Each quote is a real example of a response to the survey question.

stressed juniors who will make fewer mistakes and will have more resilience, higher productivity and less absenteeism.[10 11] This in turn will permit better patient care and more meaningful learning to take place in line with the new General Medical Council Standards for Excellence.[12] Communication and a sense of belonging to 'the team' would alleviate such perceptions of lack of support. Both consultants and specialty trainees might consider a 'mid-day catch-up' with their juniors to provide advice and moral support, phoning the junior doctor in between cases could provide an alternative if seniors were unable to leave theatre to meet at mid-day.

Improving overall support for all trainees is clearly a major factor underpinning both a better working environment but also improved patient safety. This requires a significant change in service delivery so that consultants and specialty trainees are free from all elective activity when 'on-call' and can therefore become more involved during the day in the running of the surgical wards and supporting the more junior doctors and other members of the surgical team. Many units now do offer this service and have found that this has allowed a partial return to the old 'firm' structure where the same team looks after a group of patients for several days in a row. This requires a significant change in how units and hospitals function, reorganisation of clinics, operating sessions and ward rounds, as well as identification of those surgeons who are not in theatre and might therefore be available to help out with problems on the wards when they arise during the day. The introduction of experienced 'surgical nurse practitioners' has greatly facilitated such activities, providing someone who knows the unit system, where senior staff can be found on a daily basis, as well as the unit protocols for management of specific conditions. While such persons require additional funding, those units who have invested in them have reaped rewards in improved efficiency, patient care and trainee support. In some specialties such as urology, experienced nurses can provide front line urgent clinics, thereby reducing admissions and workload for the trainee doctors.[13] Similarly, consultant led emergency 'hot clinics' have been shown to result in a significant reduction in emergency admissions and hospital stay, which in turn has an effect on reducing overall workload perception.[14]

### The challenge of working hours: safety, service and time for training

Interestingly, there were very few comments on the number of hours worked. However, linked to the numerous responses surrounding the need for more staff lies the challenge of creating rotas that comply with the European Working Time Regulations (EWTR) while at the same time facilitate excellence in training and safe surgical care. A key document published in March of 2014 highlighted the impact of the EWTR on the NHS and Health Professionals.[15] As a piece of health and safety legislation, the EWTR is intended to reduce fatigue in doctors and improve both their own safety and that of

their patients.[16] However, some professional organisations, recognise that it has had consequences, the most important of which is the reduction in the available hours for training within 'craft' specialities such as surgery.[17] It is also possible that as a measure to help combat fatigue, the necessary introduction of shift patterns may also have had an unintended consequence on patient care related to a reduction in continuity of care. Training programmes find it hard to provide the continuous rest required for the EWTR without using full shift rotas. This means that fewer trainees are available but are much busier and have to look after larger numbers of patients. This is true in both acute surgical care environments and acute medicine, where the on-call registrar is under constant pressure, another contribution to recruitment difficulties.

Finding the balance between service provision and training has been an interminable challenge for the NHS. Innovative solutions have been suggested and the authors point to the highly successful report commissioned by Health Education England—*Better Training, Better Care*[18]. This programme aimed to improve the quality of training by enabling the key recommendations from Sir John Temple's *Time for Training*[19] and Professor John Collins' *Foundation for Excellence* reports.[20] A pilot project[21] was run in Leeds and York that modified the rotas to maximise the potential time for training, and '*100% of the trainees at Leeds and York have reported more confidence in their work since the pilot and 83% agree, or strongly agree, that they have benefitted from the change in rota. The pilot saw an increase in productivity with weekday activity increasing by 37.7%, weekend activity rising by 29.1% and night shift activity by 22.1%*'.

Overall, while it appears clear that patient safety is improved by ensuring that doctors do not work excessive hours over the working week,[22] the current issues relate to arduous night shifts, often with inadequate senior support, that bring their own health-related problems. Studies have shown that staff working night shifts have an increased risk of a road traffic incident while driving home,[23] as well as less immediate increased risk of type 2 diabetes, cardiovascular disease and some cancers.[24] Other studies have looked into decision making in the context of fatigue, with health workers more likely to make mistakes when sleep deprived,[25] therefore having a direct impact on patient safety.[26]

### Building a robust surgical workforce for the future

Following the controversial introduction of the new junior doctors contract in 2016, progression into higher training is at an all-time low.[27] A survey in 2017 by the UK Foundation Programme Office found only 42.6% of foundation year 2 doctors planned to go directly into training.[28] In the light of the result of our survey, building a resilient workforce must be an absolute priority for the future well-being of the NHS. This may be combatted by diversifying our care-provider model to involve the extended surgical team,[29] with appropriate skill mix to supplement junior doctors, particularly during peaks of service demands and in areas with chronic shortages in junior medical staffing,

allowing junior doctors the time to have more meaningful patient encounters and reduce unnecessary and stressful interruptions in tasks.

### Hospital facilities

This study has shown that there remains inadequate access to hot food and appropriate facilities where staff can relax during their breaks (if they are fortunate enough to get them) without meeting patients or their relatives. Lack of a common area, such as a hospital mess, makes it more difficult for consultants and trainees to have 'catch up meetings', where advice and moral support can be provided. Improvement in the hospital working environment has been shown to improve safety and quality in hospital care while simultaneous increasing patient satisfaction.[30]

### Study strength and limitations

This study focuses on the opinions of NHS staff at the front line of delivering safe surgical care, who detail the primary problems as well as potential solutions to improve care. The survey was widely distributed throughout the UK through utilisation of many sources. Response rates across grades alleviates against subgroup bias and provides a broad spectrum of opinions. While some groups had higher response rates than others this may be explained by varying degrees of penetration and distribution across the electronic mailing lists. Despite wide distribution of the survey, the incompletion rate was higher than expected (46%). Incompletion of the survey was evenly distributed throughout the demographics; therefore, there was no indication of potential completion bias of the respondents. Survey design may have been a contributing factor to a low completion rate. The survey may have appeared as a simple tick-box exercise initially, and responders may have felt five free-text answers required more time and consideration than they were expecting (especially if completing the survey at work where there are a multitude of distractions). In addition, it may have appeared that it was compulsory to complete all five answers, which it was not, and therefore some responders may have struggled to think of five appropriate answers and terminated the survey for that reason.

Response rates were very limited from nurse practitioners, which leads to the suggestion that distribution to this group was less successful than other grades. This may have been improved by having a nurse practitioner involved in the project—as part of the SLWG—who could then have acted as a representative for nurse practitioners. The concept of the extended surgical team, which includes advanced nurse practitioners, specialty doctors, surgical first assistants, alongside surgeons and surgeons-in-training, is thankfully gaining traction within the UK, and greater efforts in the future need to be devoted to ascertaining what the many, not the few, think. Historically, doctors and nurses have operated in silos (due to dogma, governance and credentialing) but initiatives such as RCSEd's Faculty of Perioperative Care should allow the creation of a true multidisciplinary team.

Future work could also involve creating focus groups that connect with specific arms of government to help inform and influence policy makers working in areas such as workforce well-being and recruitment and retention.

## CONCLUSIONS

The results of the survey were used as a springboard for the development of a RCSEd discussion paper,[8] which lists seven key recommendations (figure 2). Our findings, alongside these recommendations, form a clear blueprint for government, policy makers and NHS that could help transform the working environment to improve patient safety, staff morale, and recruitment and retention. To those working in a significant proportion of surgical units, many of these recommendations will not seem surprising. Importantly, this survey provides qualitative evidence about what front line staff feel are the most pressing issues that need to be addressed in order to improve the working environment and create a healthy surgical workforce for the future.

**Acknowledgements** The authors are grateful to the members of the short life working group (SLWG) and Patient Safety Board (PSB) for their support and involvement in this project. Short Life Working Group members: Dr R Aloumi, Ms A Baggaley, Professor S M Griffin, Mr C McIlhenny, Mr S Paterson-Brown, Ms V Dobie, Ms A Hartley, Professor J Hill, Mr R McGregor, Professor R W Parks and Dr L Robb. Patient Safety Board members: Mr S Paterson-Brown, Ms L Ferguson, Mr T O'Brien, Mr D McArthur, Mr C McIlhenny, Mr A Geraghty, Dr N Maran, Mr G Sunderland, Professor G G Youngson, Professor K Walker, Professor P Peattie and Dr S Russ.

**Contributors** Survey distributed by all authors. AB and LR collected the data. AB and LR analysed the data and allocated prospective coding to each response. All authors were responsible for compiling the manuscript and approving the final article. AB and LR contributed equally and therefore are joint first authors.

**Funding** No external grants for used for this project, however, meeting space and travel expenses for the members of the short life working group were paid for by the Royal College of Surgeons of Edinburgh (RCSEd)

**Competing interests** SP-B and RJM were Royal College of Surgeons of Edinburgh (RCSEd) Council Members at the time of the study. SP-B is chair of RCSEd Patient Safety Board at the time of writing.

**Patient consent for publication** Not required.

**Provenance and peer review** Not commissioned; externally peer reviewed.

**Data sharing statement** Respondent-level data are available from the corresponding author at richard.mcgregor@ed.ac.uk. The presented data are anonymised with low risk of identification.

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
