## [Reviewer comments · BMJ Open]

ARTICLE DETAILS

TITLE (PROVISIONAL)	Improving the working environment for the delivery of safe surgical care in the United Kingdom: A qualitative cross-sectional analysis
AUTHORS	Baggaley, Alice; Robb, Lydia; Paterson-Brown, Simon; McGregor, Richard

VERSION 1 – REVIEW

REVIEWER	Jed Duff University of Newcastle, Australia
REVIEW RETURNED	13-Aug-2018

GENERAL COMMENTS	Your study provides some valuable insights. I provide the following comments to strengthen the manuscript: 1) The introduction could be organised more clearly. What is the context?; What is your purpose?; What is the rationale?2) There are a number of unsupported assertions in the introduction and results section. This material would be better placed in the discussion section.3) There is reference to, and results from, a pilot program in the introduction but little context about the project, particularly for the international reader.4) More description on the analysis approach is needed.5) More description of the method is needed. Validation was mentioned in the abstract but not explained in the methods. What was the inclusion/ exclusion criteria? How and where was the survey distributed? Were there reminders/follow-up? Was it anonymous?6) It's unclear what the heading, 'Patient Involvement', is for. The text that follows seems unrelated.7) Was ethics approval obtained?8) What was the response rate? What was the potential population targeted by the survey?9) Why exclude the 6 nurse practitioner responses?10) Which category of response does Box 1 represent? It would be good to have an example of responses for each category.11) More context and explanation is needed throughout the manuscript for the international reader.12) It is unclear if the recommendations in the discussion come from the participants or are from the authors.13) A table would be better than individual pie charts for comparing responses between categories of respondents.14) The limitation section in the summary is not included in the main documents.
---

REVIEWER	Rachael Lear St. George's University of London & Kingston University, UK
REVIEW RETURNED	16-Aug-2018

GENERAL COMMENTS	Abstract:  - Suggest clarifying that the questionnaire consisted of one single open-ended question invited 5 open-ended responses. - The abstract mentions 9 categories of questionnaire responses - can these be stated succinctly in the abstract? How do the 'common themes' stated in the abstract relate to the 9 categories? - "Many responses were interconnected" - unclear what this means. Introduction:  - Page 4, lines 16-17: suggest removing results from your introduction - Pages 4-5, lines 53 & 3-5: why have you include an example from medicine when your study focuses on acute surgical care? - The first part of the introduction focuses on the problems generated by the EWTR and training issues for surgeons. This detracts from the subject of the study, which is explores broader problems in the surgical environment. Human factors science is only mentioned briefly on page 5, 36. Would it not be better to open your introduction by defining Human Factors science - pointing out that this approach takes a broad view of the factors contributing to surgical safety before moving onto highlighting more specific issues, such as inadequate training, non-technical skills and team performance etc. You also mention the link between human error and patient safety. The whole ethos underpinning the Human Factors approach and the 'systems approach' is that health professionals do not make errors in isolation; our actions and behaviours are influenced by the quality of the working environment - this is an important point to highlight to justify the need for your study. Methods  - page 6, line 45 - the aim was to elicit responses from the multi-disciplinary team and patients, yet questionnaire administration seems to focus on groups with surgeon/trainee membership. Were groups with nursing or patient membership targeted? (You seem to suggest that they were in the results section - but please state in the methods section). - were there any eligibility criteria for survey completion? were any survey respondents excluded from the analysis - if so, for what reasons? - were codes specified a priori or were they generated from the data? - are the SLWG members who participated in data analysis clinicians or researchers, or other? Were they experienced in/ receive training in qualitative research/ thematic analysis? - did the independent SLWG members agree on coding of the data? if not, how were the final codes decided upon? - page 7, lines 14 - 17: 'distribution of responses within each category' - please elaborate Results  - Table 1: please give the columns headings. Please add the number of survey respondents in each group (i.e. consultant: n=? etc.) Please clarify what you mean by 'staff' (i.e. increased staffing levels??). Please also clarify what you mean by 'ring-fenced patients' - please add exact figures in brackets for the number of responses/comments provided in relation to all of the themes
--

	mentioned in the result section (i.e. "better information technology systems (n= ? responses) etc.)  - page 10, line 5 "less focus on service provision" - what do you mean by this? -page 10, lines 9-10 "only the foundation year doctors mentioned the latter suggestion in any great number." This statement is vague and it is unclear why it is included under the sub-heading 'consultants'. - page 11, line 44 - unclear what the percentages refer to Box 1:  - good to have illustrative quotes included in the results section - but perhaps add the theme they relate to, and the grade of clinician who gave that response in brackets after each quote to provide some context. The data in figures 4, 5 & 6 would be better represented in a table to allow the reader to compare percentages across clinician grade. Discussion  - "many responses were interconnected": what do you mean by this? - page 15, lines 23 - 27. Not all staff commute by car! "Many reports of fatalities due to tiredness" - really? Suggest replacing this statement with a reference that shows an association between staff fatigue and problems with patient safety (e.g. increased medication errors). - page 15, paragraph entitled 'A Focus on Human Factors' appears to be more about junior doctors training than human factors. Either change the title of the paragraph or change the content of the paragraph! Overall, the problems/targets for improvements that you have identified need to be more clearly linked with failures in patient safety in your discussion. Suggest referencing studies that have identified an association between these issues and poorer patient outcomes.
--	---

VERSION 1 – AUTHOR RESPONSE

Detailed Responses to Reviewers

Referee #1

Your study provides some valuable insights. I provide the following comments to strengthen the manuscript:

Major Comments:

- 1) The introduction could be organised more clearly. What is the context?; What is your purpose?; What is the rationale? There is reference to, and results from, a pilot program in the introduction but little context about the project, particularly for the international reader. The introduction has now been restructured to provide adequate background on the issues addressed by this project. The need for clarification of the context and purpose for this project has been addressed with the addition of the final paragraph within the introduction. Within the paragraph it has been attempted to outline the aims of the SLWG, and how these aims were implemented into the survey in order to gather 'themes'.

- 2) More description of the method is needed. Validation was mentioned in the abstract but not explained in the methods. What was the inclusion/ exclusion criteria? How and where was the survey distributed? Were there reminders/follow-up? Was it anonymous?
What was the potential population targeted by the survey?
More description on the analysis approach is needed.
Was ethics approval obtained?
The methods section of the manuscript has had significantly more substance added, with the intention for many of the points highlighted above have been addressed. Detail has been added to explain exclusion criteria in depth, as well as detailed outline of how and where the survey was distributed. The survey design has been discussed, and areas of this design have been highlighted further in discussion of the project strengths and weaknesses. The study received ethical approval from the patient safety board.
- 3) It's unclear what the heading, 'Patient Involvement', is for. The text that follows seems unrelated.
Thank you for highlighting this error in the manuscript, which hopefully has been rectified with the following paragraph:
The questionnaire used in this study was approved by the RCSEd Patient Safety Board to be distributed to patients and medical professionals. Members of the Patient Safety board are included in the acknowledgments section. The questionnaire was distributed to patient groups involved with the RCSEd, however no responses returned.
- 4) Why exclude the 6 nurse practitioner responses?
This group were excluded from initial data analysis primarily due to the very few responses when in comparison to other groups (only 6 NP). The data for this group is still available for further analysis. The reasoning for poor response from this group has been addressed in the weaknesses of the project.
- 5) Which category of response does Box 1 represent? It would be good to have an example of responses for each category
We have divided the quotes into their relevant categories.
- 6) It is unclear if the recommendations in the discussion come from the participants or are from the authors.
Please see a revised area within conclusions to clarify that the recommendations were created by the RCSEd. These recommendations were decided upon following review of the results from this study with a focus group
- 7) The limitation section in the summary is not included in the main documents.
Thank you for identifying this area lacking from the original manuscript. There is now a separate section to explore both the study strengths and weaknesses.
- 8) A table would be better than individual pie charts for comparing responses between categories of respondents.
We have converted the pie charts for the individual grades into a table for ease of comparison, but left a pie chart to depict the overall number of responses corresponding to each over-arching category.
- 9) More context and explanation is needed throughout the manuscript for the international reader.

The authors would like to thank the reviewer for identifying this within our manuscript and hope that the authors efforts can be identified through the text to create international relevance.

Referee #2

Major Comments:

Abstract:

- Suggest clarifying that the questionnaire consisted of one single open-ended question invited 5 open-ended responses. Done
- The abstract mentions 9 categories of questionnaire responses - can these be stated succinctly in the abstract? How do the 'common themes' stated in the abstract relate to the 9 categories? Done
- "Many responses were interconnected" - unclear what this means. Sentence adjusted for clarification and expanded in discussion section.

Introduction:

- Page 4, lines 16-17: suggest removing results from your introduction Done
- Pages 4-5, lines 53 & 3-5: why have you include an example from medicine when your study focuses on acute surgical care? We feel that the pressures of full shift working in the context of rota gaps is similar for both medical and surgical teams, but we have re-phrased the example to make this clearer.
- The first part of the introduction focuses on the problems generated by the EWTR and training issues for surgeons. This detracts from the subject of the study, which is explores broader problems in the surgical environment. Human factors science is only mentioned briefly on page 5, 36. Would it not be better to open your introduction by defining Human Factors science - pointing out that this approach takes a broad view of the factors contributing to surgical safety before moving onto highlighting more specific issues, such as inadequate training, non-technical skills and team performance etc. Thank you for this insight and we have moved the section regarding EWTR and added depth to our discussion regarding human factors, which we feel has provided much better context to the study.

You also mention the link between human error and patient safety. The whole ethos underpinning the Human Factors approach and the 'systems approach' is that health professionals do not make errors in isolation; our actions and behaviours are influenced by the quality of the working environment - this is an important point to highlight to justify the need for your study. We have amended this section to ensure we are clear about the ethos of human factors and not to place blame on individual clinicians.

Methods

- page 6, line 45 - the aim was to elicit responses from the multi-disciplinary team and patients, yet questionnaire administration seems to focus on groups with surgeon/trainee membership. Were groups with nursing or patient membership targeted? (You seem to suggest that they were in the results section - but please state in the methods section). We have added clarification with regards to this: we note in our limitations that we did not achieve this multi-disciplinary response.

- were there any eligibility criteria for survey completion? were any survey respondents excluded from the analysis - if so, for what reasons? Clarified. Eligibility: must have completed all the mandatory demographics questions and then provided at least one free text response. Excluded if no free text response was provided.

- were codes specified a priori or were they generated from the data? Codes were created as necessary as we went through the comments. Example provided in the methods section.

- are the SLWG members who participated in data analysis clinicians or researchers, or other? Were they experienced in/ receive training in qualitative research/ thematic analysis? SLWG all working clinicians of different grades (Consultant to F2/CT), with experience in clinical research and analysis.

- did the independent SLWG members agree on coding of the data? if not, how were the final codes decided upon? Two members worked through the >2200 responses and created the codes, discussing each new code as they came up. 92 codes were created and these were discussed with the whole of the SLWG; these codes were then grouped into themes which became the 9 categories. We hope this has been made clear in the Methods section following our amendments.

- page 7, lines 14 - 17: 'distribution of responses within each category' - please elaborate done

Results

- Table 1: please give the columns headings. Please add the number of survey respondents in each group (i.e. consultant: n=? etc.) Please clarify what you mean by 'staff' (i.e. increased staffing levels??). Please also clarify what you mean by 'ring-fenced patients' Done

- please add exact figures in brackets for the number of responses/comments provided in relation to all of the themes mentioned in the result section (i.e. "better information technology systems (n= ? responses) etc.) Done

- page 10, line 5 "less focus on service provision" - what do you mean by this? This was the statement used by the responders, not our interpretation

-page 10, lines 9-10 "only the foundation year doctors mentioned the latter suggestion in any great number." This statement is vague and it is unclear why it is included under the sub-heading 'consultants'. Amended

- page 11, line 44 - unclear what the percentages refer to Clarified

Box 1:

- good to have illustrative quotes included in the results section - but perhaps add the theme they relate to, and the grade of clinician who gave that response in brackets after each quote to provide some context. We have grouped the comments with regards to their theme.

The data in figures 4, 5 & 6 would be better represented in a table to allow the reader to compare percentages across clinician grade. Done

Discussion

- "many responses were interconnected": what do you mean by this? Clarified

- page 15, lines 23 - 27. Not all staff commute by car! "Many reports of fatalities due to tiredness" - really? Suggest replacing this statement with a reference that shows an association between staff fatigue and problems with patient safety (e.g. increased medication errors). We have expanded this section to include other risks of full shift/night shift work and included references to studies demonstrating increased risks of RTCs.

- page 15, paragraph entitled 'A Focus on Human Factors' appears to be more about junior doctors training than human factors. Either change the title of the paragraph or change the content of the paragraph! Acknowledged, and title changed.

Overall, the problems/targets for improvements that you have identified need to be more clearly linked with failures in patient safety in your discussion. Suggest referencing studies that have identified an association between these issues and poorer patient outcomes. Amended and references added.

Editorial Comments and Requests

- 1) Please revise the title to make it clear this is a qualitative study.
The title has been revised and altered to include the word qualitative in order to clarify this
- 2) The patient involvement statement could be better. Were patients or the public involved in the study design and conception? If not then please state this.
Patient involvement statement has been revised; further to this the authors have stated the members of the patient safety board at the RCSed.
- 3) The methods section is quite thin. Can this be more detailed? The methodology should be reported in enough detail for others to reproduce your study.
As stated in reference to previous reviewers' comments, the methodology section of this manuscript has been revised. The authors have provided significantly more detail of the methodology from survey design, distribution and analysis, and now feel confident that the study could be reproduced from the amount of detail provided.
- 4) You describe the completion rate but what about response rates? Why was the completion rate so low?
Further discussion points have been addressed in regard to the low completion rates. This can be found within the strengths and weaknesses sections of the revised manuscript.
- 5) There doesn't appear to be an in-depth discussion of the study's strengths and weaknesses in the discussion. Please provide one. The discussion should broadly cover the areas recommended in our Instructions for Authors: a statement of the principal findings; strengths and weaknesses of the study; strengths and weaknesses in relation to other studies, discussing important differences in results; the meaning of the study: possible explanations and implications for clinicians and policymakers; and unanswered questions and future research (http://bmjopen.bmj.com/pages/authors/#research_articles)
A more in depth discussion has been added.